

Performance evaluation of the national Norwegian early warning system for weather-
induced landslides
**Authors**
Piciullo Luca[1], Dahl Mads-Peter [2], Devoli Graziella [2,3], Colleuille Hervé[2], Calvello Michele [1]
[1] Department of Civil Engineering, University of Salerno, Italy
[2] Norwegian Water Resources and Energy Directorate, Oslo, Norway
[3] Department of Geosciences, University of Oslo, Oslo, Norway

**Abstract**
The Norwegian national landslide early warning system (LEWS), operational since 2013, is
managed by the Norwegian Water Resources and Energy Directorate and has been designed for
monitoring and forecasting the hydro-meteorological conditions potentially triggering slope
failures. Decision-making in the EWS is based upon hazard threshold levels, hydro-meteorological
and real-time landslide observations as well as on landslide inventory and susceptibility maps. In
the development phase of the EWS, hazard threshold levels have been obtained through statistical
analyses of historical landslides and modelled hydro-meteorological parameters. Daily hydro-
meteorological conditions such as rainfall, snowmelt, runoff, soil saturation, groundwater level and
frost depth have been derived from a distributed version of the hydrological HBV-model. Two
different landslide susceptibility maps are used as supportive data in deciding daily warning levels.
Daily alerts are issued throughout the country considering variable warning zones. Warnings are
issued once per day for the following 3 days with the possibility to update them according to the
information gathered by the monitoring network. The performance of the LEWS operational in
Norway has been evaluated applying the EDuMaP method, which is based on the computation of a
duration matrix relating landslide and warning events. This method has been principally employed
to analyse the performance of regional early warning model considering fixed warning zones for
issuing alerts. The original approach proposed herein allows the computation of the elements of the
duration matrix in the case of early warning models issuing alerts on variable warning zones. The
approach has been used to evaluate the warnings issued in Western Norway, in the period 2013-
2014, considering two datasets of landslides. The results indicate that the landslide datasets do not
significantly influence the performance evaluation, although a slightly better performance is
registered for the smallest and more accurate dataset. Different performance results are observed as



a function of the values adopted for one of the most important input parameters of EDuMaP, the
landslide density criterion (i.e. setting the thresholds to differentiate among classes of landslide
events). To investigate this issue, a parametric analysis has been conducted; the results of the
analysis show significant differences among computed performances when absolute or relative
landslide density criteria are considered.
**Keywords**: EDuMaP method, rainfall-induced landslides, warning zones, alert, landslide density.

## 1. Introduction

In the last decades, natural hazards caused an increased number of consequences in terms of
economic losses (Barredo, 2009) and fatalities throughout Europe (European Environment Agency,
2010; CRED, 2011). Most natural disasters are related to extreme rainfall events, which are
increasing with climate change (Easterling et al., 2000; Morss et al., 2011). The European
Commission, following an increase in human and economic losses due to natural hazards,
developed legal frameworks such as the Water Framework Directive 2000/60/EC (2000) and the
Floods Directive 2007/60/EC (2007), to increase prevention, preparedness, protection and response
to such events and to promote research and acceptance of risk prevention measures within the
society (Alfieri et al., 2012). Among the many mitigation measures available for reducing the risk to
life related to natural hazards, early warning systems (EWSs) constitute a significant option
available to authorities in charge of risk management and governance.
Within the landslide risk management framework proposed by Fell et al. (2005), landslide EWSs
may be considered a non-structural passive mitigation option to be employed in areas where risk,
occasionally, rises above previously defined acceptability levels. According to Glade and Nadim
(2014), the installation of an EWS is often a cost-effective risk mitigation measure and in some
instances the only suitable option for sustainable management of disaster risks. Rainfall-induced
warning systems for landslides are, by far, the most diffuse class of landslide EWS operating
around the world. Two categories of landslide EWSs can be defined on the basis of their scale of
analysis: "local" and "regional" systems (ICG 2012; Thiebes et al. 2012; Calvello et al. 2015, Stähli
et al., 2015). Regional landslide EWSs for rainfall-induced landslides have become a sustainable
risk management approach worldwide to assess the probability of occurrence of landslides over
appropriately-defined wide warning zones. In fact during the last decades, several systems have
been designed and improved, not only in developing countries (UNISDR 2006; Chen et al., 2007;
Huggel et al., 2010; among others) but also in developed countries (NOAA-USGS, 2005; Badoux et
al., 2009; Baum and Godt, 2010; Osanai et al., 2010; Lagomarsino et al., 2013; Tiranti and



Rabuffetti, 2010; Rossi et al., 2012; Staley et al., 2013; Calvello et al., 2015; Segoni et al., 2015).
As a recent example, the Norwegian landslide EWS was launched in autumn 2013 by the
Norwegian Water Resources and Energy Directorate (NVE). The regional system has been
developed for monitoring and forecasting the hydro-meteorological conditions triggering landslides
and to inform local emergency authorities in advance about the occurrence of possible events
(Devoli et al., 2014). Daily alerts are issued throughout the country in variable warning zones. The
evaluation of the alerts issued, i.e., the performance of the early warning model that comprises the
EWS (Calvello and Piciullo, 2016), is not a trivial issue, and regular system testing and
performance assessments (Hyogo Framework for Action, 2005) are fundamental steps. The
performance analysis can be an awkward process because some important aspects can be sparsely
evaluated. The EDuMaP method (Calvello and Piciullo, 2016) can be seen as a powerful tool to
help system managers and researchers in the performance evaluation of regional warning models.
Up to now, this method has been applied exclusively to evaluate the performance of regional
warning models designed for issuing alerts in fixed warning zones (Calvello and Piciullo, 2016;
Piciullo et al., 2016a,b; Calvello et al., 2016). In the present study the EDuMaP method has been
adapted to evaluate the performance of the alerts issued in variable warning zone. Moreover, the
procedure has been tested on the Norwegian landslide EWS in the period 2013-2014.

## 2. The national landslide early warning system for rainfall-and snowmelt-induced landslides in Norway

### 2.1  Physical setting

Norway covers an area of ~ 324,000 km$^2$. With its elongated shape of 1800 km, the country reaches
from latitude 58°N to 71°N. Approximately 30% of the land area are mountainous, with the highest
peaks reaching up to 2500 m. a.s.l and slope angles over 30 degrees covering 6,7% of the country
(Jaedicke *et al.,* 2009). In geological terms, Norway is located along the western margin of the
Baltic shield with a cover of Caledonian nappes in the western parts of the country (Etzelmüller *et*
*al.,* 2007; Ramberg *et al.,* 2008). The Caledonian nappes are dominated by Precambrian rocks and
metamorphic Cambro-Silurian sediments, while the bedrock in the Baltic shield is dominated by
Precambrian basement rocks. Cambro-Silurian sediments and Permian volcanic rocks are found in
the Oslo Graben (Ramberg *et al.,* 2008).
Recurrent glaciations, variations in sea level and land subsidence/uplift, as well as weathering,
transport and deposition processes have created the modern Norwegian landscape (Gjessing, 1978;



Ramberg *et al.,* 2008). Thus, dominating quaternary deposits include various shallow (in places
colluvial) soils, as well as moraine and marine deposits, (**Fig. 1**).
Because of the latitudinal elongation and the varied topography, the Norwegian climate displays
large variations. Along the Atlantic coast, the North Atlantic Current influences the climate whereas
the inland areas experiences a more continental climate. Based on the Köppen classification
scheme, the Norwegian climate can be classified in three main types: warm temperate humid
climate, cold temperate humid climate and polar climate (Gjessing, 1977). Precipitation types can
be divided into three categories: frontal, orographic and showery. The largest annual precipitation
values are found near the coast of Western Norway (herein also called Vestlandet) with up to 3575
mm/year. In contrary, the driest areas receiving <500 mm/year are found in parts of South-Eastern
Norway (Østlandet) and Finnmark county (Førland, 1993).

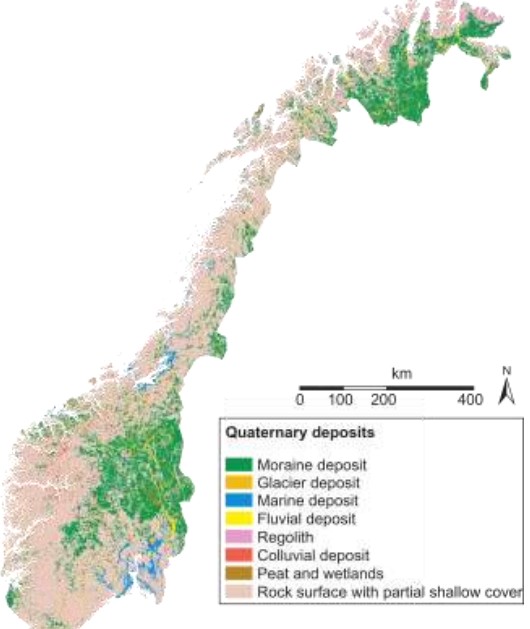

**Fig. 1**. Overview of quaternary deposits in Norway. Modified from NGU, (2012).

Steep landforms in combination with various soil and climatic properties provide a basis for several
types of shallow landslides in non-rock materials. These slope failures include slides in various
materials, debris avalanches, debris flows and slush flows. Landslides are mostly triggered by
rainfall, often in combination with snowmelt. Some events are also triggered from/initiated as
rockfall or slush flows, developing into, for example, debris flows as they propagate downslope.



Shallow landslides constitute a substantial threat to the Norwegian society. According to Furseth
(2006), at least 230 people have been killed by such slope failures during the latest approximately
500 years. In the period 2000-2009, road authorities registered more than 1800 shallow landslides
along Norwegian roads (Bjordal & Helle, 2011).

**2.2 The national landslide early warning system**
In order to mitigate the risk from shallow landslides, a national EWS has been developed at the
Norwegian Water Resources and Energy Directorate (NVE) as part of the national responsibility on
landslide risk management. The system is established to warn about the hazard of debris flows,
debris slides, debris avalanches and slush flows at regional scale. The EWS, operative since 2013,
has been developed in cooperation with the Norwegian Meteorological Institute (MET), Norwegian
Public Road Administration (SVV) and the Norwegian National Rail Administration (JBV).

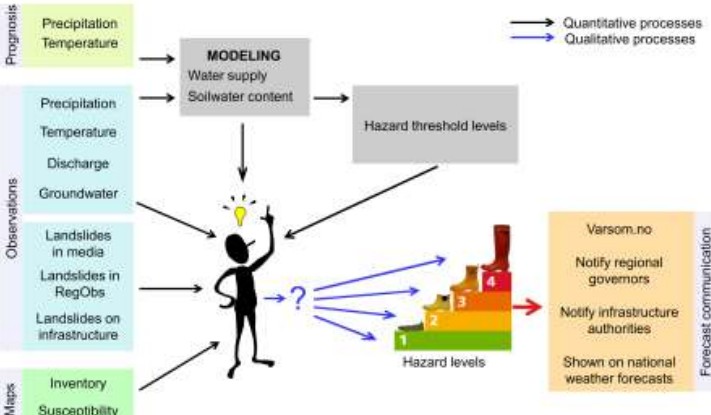


**Fig. 2**. Organization of the landslide early warning system in Norway.

Decision-making in the EWS is based upon hazard threshold levels, hydro-meteorological and real-
time landslide observations as well as landslide inventory and susceptibility maps (**Fig. 2**). In the
development phase of the EWS, hazard threshold levels have been investigated through statistical
analyses of historical landslides and modelled hydro-meteorological parameters. Daily hydro-
meteorological conditions such as rainfall, snowmelt, runoff, soil saturation, groundwater level and
frost depth have been obtained from a distributed version of the hydrological HBV-model (Beldring
*et al.,* 2003).





Hazard threshold levels presently used in the EWS were proposed by Colleuille *et al.* (2010). The
thresholds, combining simulations of relative water supply of rain or snowmelt and relative soil
saturation/groundwater conditions, were derived from empirical tree-classification using 206
landslide events from different parts of the country (**Fig. 3**). Later analyses, summarized by Boje *et*
*al.* (2014), confirm the good performance of combining soil water saturation degree and normalised
rainfall and snowmelt.

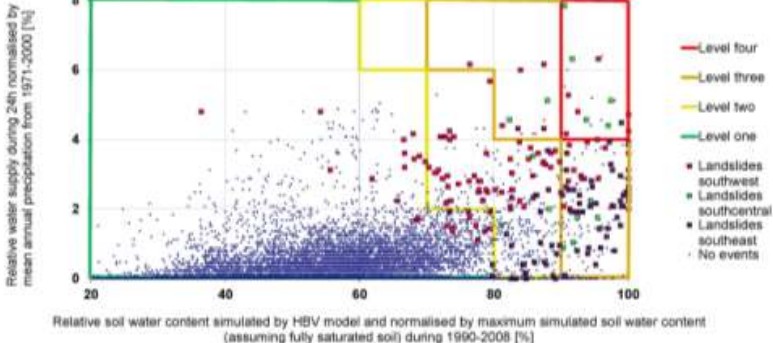

**Fig. 3**. Hydrometeorological hazard thresholds used in the Norwegian EWS.

Two different landslide susceptibility maps are used as supportive data in the process of setting
daily warning levels. One map indicates initiation and runout areas for debris flows at slope scale
(Fischer *et al.,* 2012), while another indicates susceptibility at catchment level, based upon
Generalized Additive Models (GAM) statistics (Bell *et al.,* 2014).
Susceptibility maps, hazard threshold levels and other relevant data are displayed in real-time in a
webpage, www.xgeo.no, which is used as decision expert tool to forecast various natural hazards
(floods, snow avalanches, landslides). Landslide hazard threshold levels and hydrometeorological
forecasts are displayed as raster data with 1 km$^2$ resolution, whereas susceptibility maps, landslide
information (historical and real-time) and hydrometeorological observations are shown as either
raster, polygon or point data.
A landslide expert on duty (as member of a rotation team) uses the information from forecasts,
observations, maps and uncertainty in weather forecasts to qualitatively perform a nationwide
assessment of landslide warning levels (**Fig. 2**). Four warning levels are defined: green (1), yellow
(2), orange (3), and R (4) showing the level of hazards, or more exactly the recommended
awareness level (**Tab. 1**). The warning period follows the time steps of quantitative precipitation





and temperature forecasts used to simulate other hydro-meteorological parameters, and thus lasts
from 06:00 UTC to 06:00 UTC each day. Warning levels are updated twice during the 24 hour
warning period (morning and afternoon) and are published in the webpage www.varsom.no.
Warnings at yellow, orange and R level are also sent to emergency authorities (regional
administrative offices, roads and railways authorities) and media. Warning zones are not static
geographical warning areas. Instead they vary from a small group of municipalities to several
administrative regions, depending on current hydro-meteorological conditions (**Fig. 4**). Thus, extent
and position of warning zones are dynamic and change from day to day.

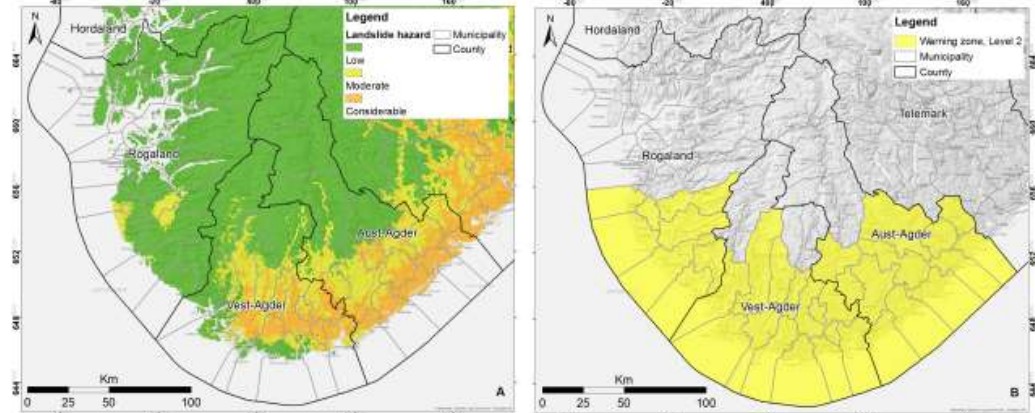


**Fig. 4**. A: Hydrometeorological thresholds indicating potential landslide hazard in the counties of
Rogaland, Vest-Agder, Aust-Agder and Telemark in South-Eastern Norway on 15.02.2014. B: The
resultant early warning zone, on warning level 2 ("yellow level") issued on 15.02.2014 for the same
area and including about 32 municipalities.

2.3   **Current performance evaluation of the EWS**
To evaluate the performance of a regional landslide early warning model, a comparison of issued
landslide warning levels and subsequent event information is carried out on a weekly basis. Event
information is reported by Roads/Railways Authorities or municipalities, as well as obtained from
media and from a real-time database to register observations. The latter has been designed as a
public tool supporting crowd sourcing (Ekker et al. 2013), and is currently available to the public as
telephone application and website at www.regobs.no. Categorization of issued warning levels into
false alarms, missed events, correct and wrong levels is based on semi-quantitative classification



criteria for each warning level (**Tab. 1**). The principle behind the criteria is that rare hydro-
meteorological conditions are expected to cause more landslides and possibly higher damages.
Thus, the criteria contain information on the expected number of landslides per area, as well as
hazard signs indicating landslide activity. As seen in **Table 1** the ranges chose for the number of
expected landslides and the size of the hazardous areas at each warning level are quite wide. This
choice is due to the fact that the EWS is relatively new and still in a phase of continuous
development.

**Tab. 1**. Criteria for evaluating daily warning levels in the Norwegian EWS.

| Warning level | Classification criteria |
|---|---|
| **4 (Red)** | > 14 landslide (per 10-15.000 km2)<br>Hazard signs: Several road blockings due to landslides or flooding |
| **3 (Orange)** | 6-10 landslides (per 10-15.000 km2)<br>Hazard signs: Several road blockings due to landslides or flooding |
| **2 (Yellow)** | 1-4 landslides (per 10-15.000 km2)<br>Hazard signs: flooding/erosion in streams |
| **1 (Green)** | No landslides<br>1-2 landslide caused by local rain showers<br>1 small debris slide if in area with no signs of elevated warning level<br>Man-made events (from e.g. leakage, deposition, construction work or explosion) |


## 203 3. Performance evaluation of the LEWS in Western Norway for the period
## 204      2013-2014

### 205 3.1 Study area and landslide data

The study area includes the four administrative regions of Møre og Romsdal, Sogn og Fjordane,
Hordaland and Rogaland located on the Norwegian west-coast. A common name for the entire area
is Vestlandet (i.e. Western Norway) (**Fig. 1**). The area is dominated by narrow fjords and steep
mountainsides reaching from sea level to 1000 m a.s.l. or more, and high annual precipitation of up
to ~3500 mm, (Førland, 1993). Shallow quaternary deposits cover locally weathered and altered
bedrock of mainly precambric and Caledonian metamorphic and magmatic origin. As a result,
Vestlandet is highly prone to landslides, in particular, debris avalanches, debris flows and slush
flows.
Vestlandet is the rainiest area of Norway with many annual precipitation episodes bringing high
amounts of rain and/or snow. Precipitation patterns and spatial distribution display large variations



within the study area. The following precipitation patterns are observed described based on the
main spatial distribution:
a) NNW precipitation only in the region of Møre og Romsdal;
b) NW precipitation mainly in the regions of More og Romsdal and Sogn og Fjordane, or
sometimes in the northern part of Hordaland;

c) WNW precipitation in the entire study area;
d) W precipitation distributed mainly in Sogn og Fjordane, Hordaland and Rogaland;
e) SW precipitation distributed mainly in Rogaland and Hordaland, or sometimes also in Sogn
of Fjordane;

f) SSW precipitation only in Rogaland, or sometimes in Hordaland and rarely in the southern
part of Sogn og Fjordane;

g) S and SE with precipitation mainly in South-Eastern Norway (in summer) and not in the
study area, however because of size of the systems, precipitation can spread to Møre og
Romsdal or to eastern Sogn og Fjordane or Hordaland, depending on trajectory;

h) Local showers (mostly in summer), with clusters of maximum precipitation distributed
randomly within the study area;

i) Southern Norway, with precipitation distributed in the entire southern part of the country
and consequently in the entire study area.

During the years 2013 and 2014 more than 70 precipitation episodes, i.e. rain and/or snow records
with more than 30 mm/24h, were registered, with some episodes bringing more than 75-150
mm/24h of rain/snow to the entire study area or part of it, following the patterns indicated above.
Duration of precipitation episodes ranged from 1 day to 14-18 consecutive days, particularly during
autumn.
Landslide early warnings higher than green level were issued for 49 days during the two-year
period (**Tab. 2**). Most of these were at yellow level, however five warnings at orange level were
issued in 2014 in 3 consecutive days. In 12 cases, the yellow warnings issued during the morning
evaluation was downgraded to green later the same day. The most significant precipitation episodes
recorded in 2013-2014 are 11 and occurred in the following days:   14-15/04/13,       12-13/08/13,

244 7/10/13, 22/10/13, 15/11/ 13, 28/12/ 13, 23/02/ 14, 20/03/14, 14/07/14, 18-19/08/14, 27-28/10/14.








**Tab. 2**. Significant rainfall, number of days with at least one warning, number of warnings and
landslides in the period 2013-2014.

|  | 2013 | 2014 | tot |
|---|---|---|---|
| **Precipitation episodes, i.e. rainfall and/or snow > 30 mm/24h** | 41 | 32 | 73 |
| **Number of days with at least one warning** | 20 | 29 | 49 |
| **Number of warnings** | 21 | 39 | 60 |
| **red warnings** | 0 | 0 | |
| **orange warnings** | 0 | 5 | |
| **yellow warnings** | 21 | 34 | |
| **Number of landslides** | 204 | 181 | 385 |


Examples of warnings issued during 2013 and 2014 are showed in **Figure 4**. Most of the alerted
warning zones  were completely included in the study area (**Fig. 5c, d, f**). However, some warnings
were mainly issued for neighboring areas, to the 4 regions chosen as case study (**Fig. 5a, b, e**). The
examples in **Fig. 5** also illustrates the diversity in having variable instead of fixed warning zones.

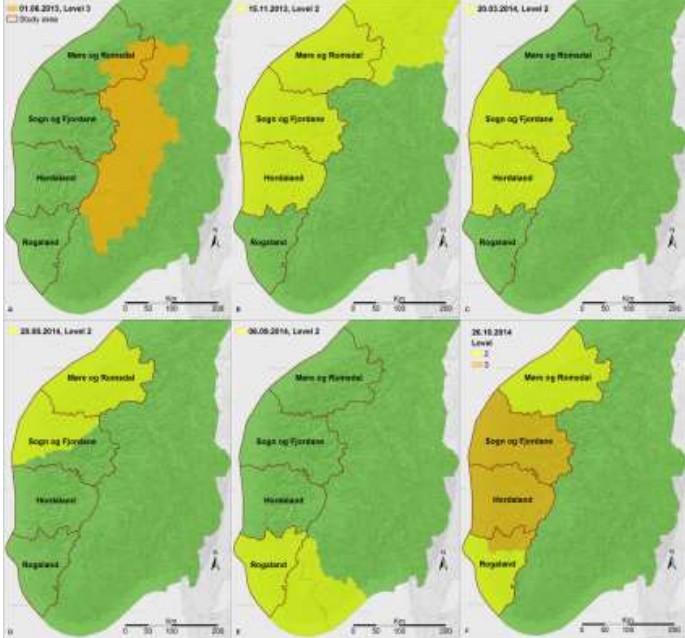

**Fig. 5**. Examples of early warning areas and levels during 2013-2014.




Within the study area, for the period 2013-2014, the Norwegian national landslide database
(www.skrednett.no) lists 476 landslides in soils and/or slush flows. Due to errors and double
registration, 385 of these slope failures were considered valid for the current analyses (**Fig. 6** and
**Tab. 3**): 65% are categorized as landslide in soil, not otherwise specified due to lack of further
documentation; 17% are categorized as debris avalanches, following Hungr et al. (2014), in many
cases initiated as small debris slides; 7% are classified as debris flows, following Hungr et al.
(2014); 5% are soil slides in artificial slopes (cuts and fillings along roads or railway lines); 5% are
slush flows and the remaining 1% are rock falls developing into debris avalanches.

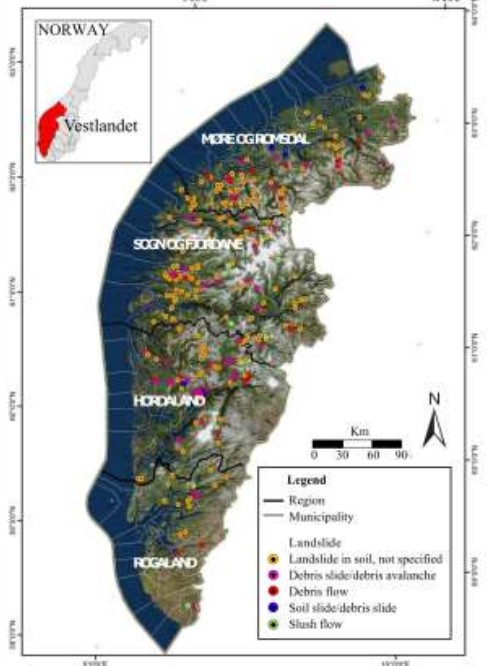

**Fig. 6**. Location and classification of landslides occurred within the study area during 2013-2014.
**Tab. 3**: Classification of landslides in soils and slush flows in the period 2013-2014.

| Landslide type | n | % |
|---|---|---|
| Landslide in soil, not specified | 249 | 65 |
| Debris slide/debris avalanches | 65 | 17 |
| Debris flows | 27 | 7 |
| Rock fall/Debris avalanches | 5 | 1 |
| Slush flows | 19 | 5 |
| Soil slide in artificial slopes | 20 | 5 |
| Total | 385 | |




The EDuMaP method was applied to two different sets of phenomena: Set A and Set B. The first set
includes all 385 slope failures, while the second included only 131 phenomena, as "landslide in soil
not specified" and "rock fall/debris avalanches" were removed from this dataset. The removal of
non-specified landslides was due to the questionable quality of these registrations in the national
landslide database, while the exclusion of rock falls inducing debris avalanches was due to
uncertainty on whether precipitation can indeed be considered their triggering cause.
## 3.2  The EDuMaP method
The paper proposes the evaluation of the performance of the landslide early warning system
operational in Norway by means of the "Event, Duration Matrix, Performance (EDuMaP) method"
(Calvello & Piciullo, 2016). This method has been principally employed to analyse the performance
of regional early warning model considering fixed warning zones for issuing alerts. The method
comprises three successive steps: identification and analysis of landslide and warning Events (E),
from available databases; definition and computation of a Duration Matrix (DuMa), and evaluation
of the early warning model Performance (P) by means of performance criteria and indicators.
The first step requires the availability of landslides and warnings databases for the preliminary
identification of "landslide events" (LEs) and "warning events" (WEs). A landslide event is defined
as one or more landslides grouped on the basis of their spatial and temporal characteristics. A
warning event is defined as a set of warning levels issued within a given warning zone, grouped
considering their temporal characteristics. The parameters which need to be defined to carry on the
events analysis are ten: 1) warning levels, $W_{lev}$; 2) landslide density criterion, $L_{den(k)}$; 3) lead time,
$t_{LEAD}$; 4) landslide typology, $L_{typ}$; 5) minimum interval between landslide events, $\Delta t_{LE}$; 6) over time,
$t_{OVER}$; 7) area of analysis, A; 8) spatial discretization adopted for warnings, $\Delta A_{(k)}$; 9) time frame of
analysis, $\Delta T$; 10) temporal discretization of analysis, $\Delta t$. For more details see Calvello and Piciullo,
2016. The second step of the method is the definition and computation of a "duration matrix",
whose elements report the time associated with the occurrence of landslide events in relation to the
occurrence of warning events, in their respective classes. The number of rows and columns of the
matrix is equal to the number of classes defined for the warning and landslide events, respectively
(**Figure 7**). The final step of the method is the evaluation of the duration matrix based on a set of
performance criteria assigning a performance meaning to the element of the matrix. Two criteria are
used for the following analyses (**Fig. 7**), respectively indicated as criterion 1 and criterion 2. The
first criterion employs an alert classification scheme derived from a 2x2 contingency table, thus
identifying: correct predictions, CPs; false alerts, FAs; missed alerts, MAs; true negatives, TNs. The



second criterion assigns a color code to the elements of the matrix in relation to their grade of
correctness, classified in four classes as follows: green, G, for the elements which are assumed to be
representative of the best model response; yellow, Y, for elements representative of minor model
errors; red, R, for elements representative of a significant model errors; purple, P, for elements
representative of the worst model errors. A number of performance indicators may be derived from
the two performance criteria described. **Table 4** reports the name, symbol, formula and value of the
performance indicators considered herein.

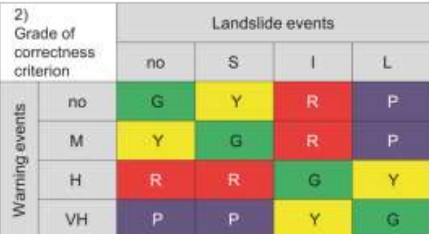

**Fig. 7**. Performance criteria used for the analyses performed herein (modified from Calvello &
Piciullo, 2016). Four classes of warning events (key: no, no warning; M, moderate warning; H, high
warning; VH, very high warning) and four classes of landslide events (key: no, no landslides; S,
small event, few landslides; I, intermediate event, several landslides; L, large events, many
landslides).
**Tab. 4**. Performance indicators used for the analysis.

| Performance indicator | Symbol | Formula |
|---|---|---|
| Efficiency index | $I_{eff}$ | $CP/\Sigma_{ij}d_{ij}$  (excluding $d_{11}$) |
| Hit rate | $HR_L$ | $CP/(CP+MA)$ |
| Predictive power | PPW | $CP/(CP+FA)$ |
| Threat score | TS | $CP/(CP+MA+FA)$ |
| Odds ratio | OR | $CP/(MA+FA)$ |
| Miss classification rate | MR | $1- I_{eff}$ |
| Missed alert rate | $R_{MA}$ | $MA/(CP+MA)$ |
| False alert rate | $R_{FA}$ | $FA/(CP+FA)$ |
| Error Rate | ER | $(Red\&Pur)/ \Sigma_{ij} d_{ij}$ (excluding $d_{11}$) |
| Missed and false alerts balance | MFB | $MA/(MA+FA)$ |
| Probability of serious mistakes | $P_{SM}$ | $Pur/\Sigma_{ij}d_{ij}$ (excluding $d_{11}$) |



### 3.3 Adaptation of the EDuMaP method to variable warning zones

In earlier studies, the EDuMaP method has been applied to analyse the performance of regional landslide EWSs adopting a fixed spatial discretization for warnings. In contrast, the Norwegian landslide EWS employs variable warning zones. This characteristic influences the first two phases of the EDuMaP method and thus requires some adaptation of the method to the current study. This section explains how to define landslide events (LEs) and warning events (WEs) and how to compute the duration matrix in case of variable warning zones.

The Norwegian EWS uses municipalities as the minimum warning territorial unit (TU). Hence, municipalities alerted with the some warning level are grouped together, defining a warning zone of level $i$ (**Fig. 5**). The considered EWS adopts four warning levels. Therefore, on each day of alert, up to four different warning levels can be issued. LEs and WEs need to be defined for each warning zone and day of alert. As seen in **figure 8**, LEs are defined by grouping together landslide occurrences within the areas alerted, i.e. warning zone, with equal warning level $i$. For instance, in Day 1 two distinct landslide events appears, containing 4 and 1 landslides, respectively. The first event belongs to the warning zone alerted with level 2 and the latter to the warning zone alerted with level 1. In Day 3 there are 4 warning zones, each one alerted with a different warning level and 4 distinct LEs can be identified, one per warning zone. The class each LE belong to, as defined in **section 3.2**, depends on the landslide density criterion, $L_{den(k)}$, chosen for the analyses.

The duration matrix is evaluated for the whole area of analysis, A, in a period of analysis, ΔT, summing the time$_{ij}$ computed within the different warning zones, for each temporal discretization Δt. In particular, the values of time$_{ij}$ are computed as follows:

$$\text{time}_{ij} = \sum\nolimits_{\Delta t} \frac{(TUA_{ij})}{A} \qquad \textbf{(Eq. 1)}$$

where: Δt is the minimum temporal discretization, in this case equal to 1 day; A is the area of analysis; $TUA_{ij}$ is the area of the territorial unit with level of the warning event, $i$, and class of the landslide event, $j$, per day of alert. Each element of the duration matrix, $d_{ij}$, is then computed, within the time frame of the analysis, ΔT, as follows:

$$d_{ij} = \sum\nolimits_{\Delta T}(\text{time}_{ij}) \qquad \textbf{(Eq. 2)}$$

This computation is herein exemplified for three hypothetical days, using a landslide density criterion, $L_{den(k)}$ in four classes. In **Figure 8**, four classes of LEs have been considered: 0 (no landslides), small (1-2 landslides), Intermediate (3-4 landslides) and Large (≥5 landslides). The hypothetical EWS in **Fig. 8** also has four warning levels, $W_{lev}$: green, yellow, orange and red. At "day 1" two different warning zones can be defined grouping together the TUs (blue boundary in



**Fig. 8**) with the same warning level. The warning zones are composed by 10 and 8 TUs, and they
are alerted with two different warning levels: green and yellow. In the two warning zones, a "small"
LE and an "Intermediate" LE, respectively, are occurred. Once the warning levels and the LEs
within each warning zone have been defined, $time_{12}$ and $time_{23}$ are evaluated for each TU using
**Equation 1**. At "day 2" three warning zones and two "Small" LEs have been identified. At "day 3"
LEs are occurred in each of the four warning zones identified. Finally, the evaluation of elements
$d_{ij}$, is carried out following **Equation 2**, over the time frame of the analysis, $\Delta T$.

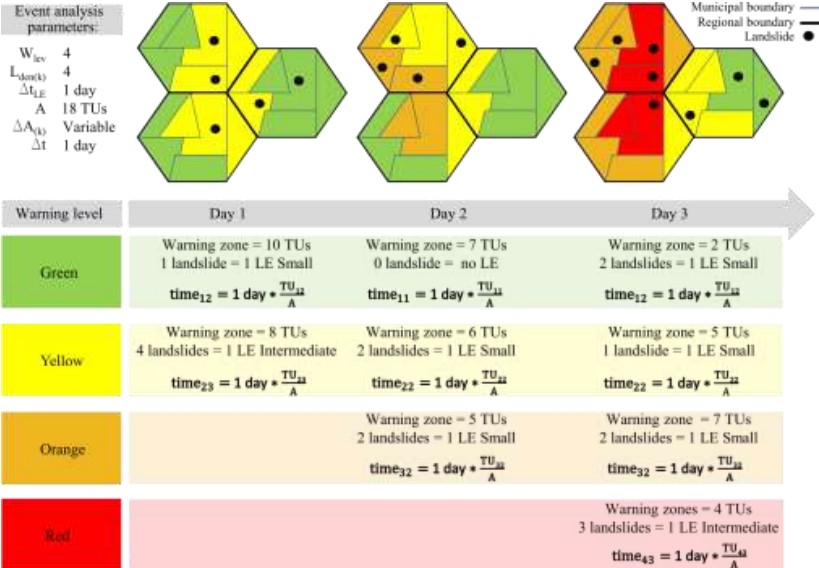

**Fig. 8**: Computation of $time_{ij}$ elements as a function of warning levels and LEs occurred for each
warning zone for three hypothetical days of warning.


# 4. Results and discussion
## 4.1  Events analysis
As previously mentioned, the events analysis phase of the EDuMaP method depends on the values
assumed by a series of well-identified parameters, which are defined to allow the analyst to make
choices on how to select and group landslides and warnings.
**Table 5** shows the values of the ten input parameters, cf. section 3, for the two analyses carried out,
i.e. case A and case B. The values are representative of the structure and operational procedures of
the warning model employed in the Norwegian EWS. The period of analysis, $\Delta T$, is 2013-2014,



while Δt, is set to 1 day. Parameters $t_{LEAD}$ and $t_{OVER}$ are both set to zero. The four warning levels,
$W_{lev}$, are: green (no warning), yellow ($WL_1$), orange ($WL_2$), red ($WL_3$). The landslides used for the
analyses are grouped into landslide events considering a $\Delta t_{LE}$ of 1 day. The four classes of LEs are
defined employing a relative landslide density criterion, $L_{den(k)}$, as a function of both number of
landslides and territorial extensions. The values have been derived by the criteria for the daily
warning levels evaluation in the Norwegian EWS (see **Tab. 1**). The only difference between case A
and case B has to do with the type of landslides used for the analyses, which respectively refer to
the datasets A and B as defined in **Table 2**.

**Tab. 5**: Values of the EDuMaP input parameters for the two analyses: case A and case B

|                | Case A | Case B |
|----------------|--------|--------|
| $W_{lev}$      | 4      | 4      |
| $L_{den(k)}$   | 4 – Relative criterion | 4 – Relative criterion |
| $t_{LEAD}$     | 0      | 0      |
| $L_{typ}$      | set A  | set B  |
| $\Delta t_{LE}$ | 12     | 12     |
| $t_{OVER}$     | 0      | 0      |
| A              | 4 Regions on the Norwegian west coast | 4 Regions on the Norwegian west coast |
| $\Delta A_{(k)}$ | Variable | Variable |
| $\Delta T$     | 2013-2014 | 2013-2014 |
| $\Delta t$     | 1 day  | 1 day  |


Dataset A is composed by 385 rainfall- and snowmelt-induced landslides occurring within the study
area. These slope failures have been grouped into 137 LEs. The majority of LEs belong to class
"Small" (133 events), while the rest of them (4 events) belong to class "Intermediate"; no "Large"
LEs have been recorded in the period of analyses (**Tab. 6**). For case B, the 131 considered
phenomena have been grouped into 57 LEs, 54 "Small" and 3 "Intermediate" events (**Tab. 6**). A
total of 60 warnings were issued in the period of analysis; none of these were "Red". Five warning
zones received the level "Orange" and 55 zones received the warning level "Yellow". In the period
of analysis 37 different warning zones have been alerted (**Tab. 6**).








**Tab. 6**: Number of landslides, landslides, warning events issued and warning zones alerted in 2013-
2014 in the area of analysis.

|  | Case A | Case B |
|---|---|---|
| Landslide | 385 | 131 |
|  |  |  |
| Landslide events, LE | 137 | 57 |
| Small | 132 | 54 |
| Intermediate | 5 | 3 |
| Large | 0 | 0 |
|  |  |  |
| Warning events, WE | 60 | 60 |
| Warning zones alerted | 37 | 37 |


## 4.2    Performance evaluation for the years 2013-2014

Two different sets of landslides have been considered in the performance of the Norwegian EWS
for the Vestlandet area: Set A and Set B. The duration matrices obtained are shown in **table 7**. Both
cases refer to the years 2013-2014, thus, the sum of matrix elements is always equal to 730 days.
**Tab. 7**: Duration matrices for cases A and B, units of time expressed in days.

| CASE A |  | LE class | | | |
|---|---|---|---|---|---|
|  |  | 1 | 2 | 3 | 4 |
|  | 1 | 600,48 | 107,62 | 0,00 | 0,00 |
| **WE level** | 2 | 9,88 | 8,47 | 1,80 | 0,00 |
|  | 3 | 0,00 | 1,16 | 0,58 | 0,00 |
|  | 4 | 0,00 | 0,00 | 0,00 | 0,00 |


| CASE B |  | LE class | | | |
|---|---|---|---|---|---|
|  |  | 1 | 2 | 3 | 4 |
|  | 1 | 671,55 | 36,56 | 0,00 | 0,00 |
| **WE level** | 2 | 11,32 | 7,90 | 0,93 | 0,00 |
|  | 3 | 1,16 | 0,00 | 0,58 | 0,00 |
|  | 4 | 0,00 | 0,00 | 0,00 | 0,00 |

The duration matrices have been analysed considering two different performance criteria (see
**Figure 6**). The first one is derived by a contingency table scheme (criterion 1), the other one is
based on a colour code assigning a grade of correctness to each matrix cell (criterion 2). The results
obtained considering criterion 1 for both Case A and B (**Fig. 9.a**) show a very high percentage of
correct predictions (CPs), over 96%, and around 1,5% of missed alerts (MAs). The amount of false




alerts (FAs) are 1% and 2% respectively for Case A and B. Following criterion 2 (**Fig. 9.b**)
differences, among Case A and B, can be observed in terms of greens (G), that are respectively
equal to 7% and 14,5%, and yellows (Y) that are respectively equal to 91% and 82%. No P and just
few R, equal to 2,3% and 3,6%, are observed in Case A and Case B, respectively. Following
criterion 1, there are not significant differences among the two cases analysed. In terms of criterion
2, Case B shows higher values of G. This means that considering the reduced set of landslides (Set
b), there is a better correspondence between the LE classes and corresponding warning levels
issued.

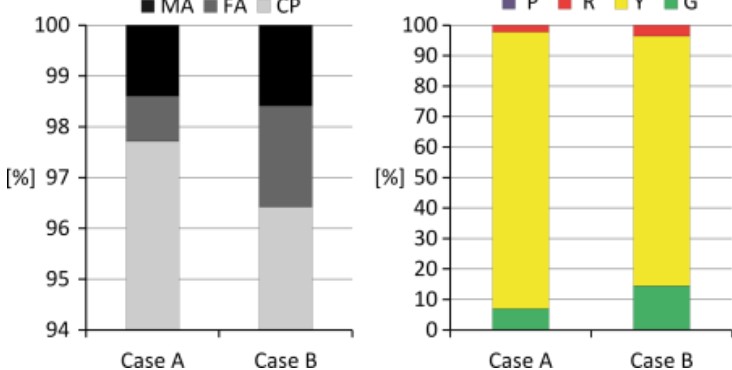

**Fig. 9**: Duration matrix results in terms of: a) criterion 1; b) criterion 2
The performance indicators used to analyse the duration matrices (**Tab. 2**) are grouped into two
subsets of indicators, respectively evaluating success and error (**Fig. 10**). Excluding the odds rate
(OR), the remaining success indicators have a percentage higher than 95% for both cases, due to the
high value of CPs that is orders of magnitude higher than MAs and FAs. Therefore the OR, that
indicates the correct predictions relative to the incorrect ones, assumes a very high value for both
cases, although slightly higher for Case A (**Fig. 11**). The error indicators MR, ER, RMA and RFA
assume very low values and the differences between the two cases are around 1% (**Fig. 10.b**). The
MFB, which  represents the ratio of MAs over the sum of MAs and FAs, is around 60% and 45%
respectively for Cases A and B (**Fig. 11**).




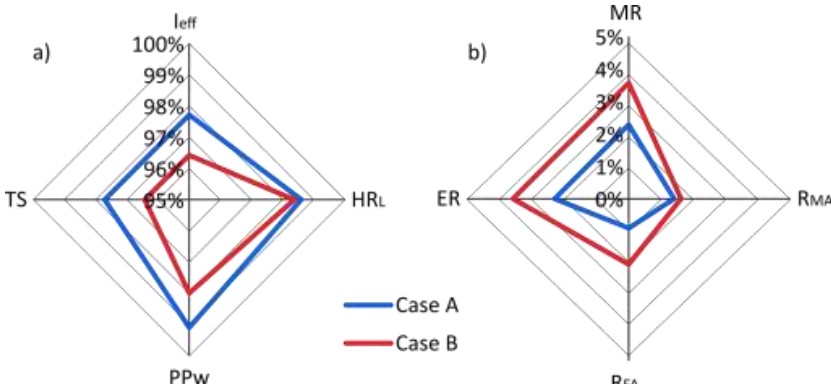

**Fig. 10**: Performance indicators quantifying the landslide early warning performance of Case A (in
blu) and Case B (in red) in terms of success (a) and error (b).

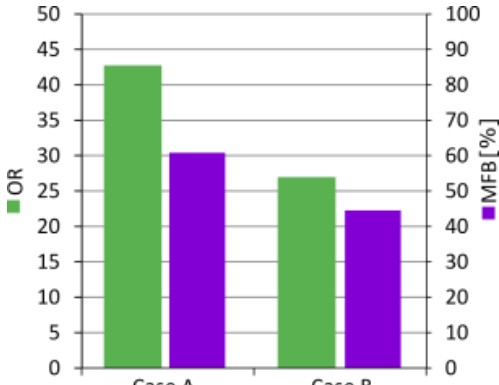

**Fig. 11**: Odds Ratio (OR) and Missed and False alerts Balance (MFB) performance indicators,
quantifying the landslide early warning performance of Case A and Case B.
In this performance analysis the high value of $I_{eff}$, (>95%) and ORs, could be interpreted as an
excellent result but, in contrast, the high value of MFB highlights some issues related to the
duration of MAs in relation to the total duration of wrong predictions. In general, this could be a
serious problem because MAs mean that no warnings or low level warnings have been issued
during the occurrence of one or more LEs of the highest two classes ("Intermediate" and "Large").
In particular for Case A, 4 out of 5 LE of class "Intermediate" have to be considered MAs because
they occurred when the warning was set to level 2. Following the previous considerations, Case B
shows the best performance in terms of both success and error indicators, with a lower value of
MFB and a high value of OR. Case B uses a landslide dataset composed of rainfall-induced
landslides with a higher accuracy of information than Case A. As stated in Piciullo et al., (2016),



the result of a performance evaluation is strictly connected to the availability of a landslide
catalogue and to the accuracy of the information included in it.
Finally, it is important to stress the use of both success and error indicators to carry out a complete
performance analysis. As in this case, dealing with some indicators neglecting others could cause a
wrong evaluation of the early warning model performance. For instance, in the period of analysis,
no LEs of class 4 and only few LEs of class 3 (see **Tab. 6**), occurred. However, the majority of
durations of these LEs have been missed (**Tab. 7**). This means that the landslide early warning
model was mostly able to predict LEs of class "Small". A possible solution to obtain a better model
performance, reducing MAs and simultaneously increasing CPs and G, could be to decrease the
thresholds employed to issue the warning level "High".

### 4.3  Parametric analysis: the landslide density criterion

A parametric analysis on the landslide density criterion, $L_{den(k)}$, has been herein conducted with  a
twofold purpose: to compare the performance of different early warning models, and to evaluate the
effect of the choices that the analyst makes when defining landslide event (LE) classes on the
performance indicators computed according to the EDuMaP method. The landslide density, $L_{den(k)}$,
represents the criterion used to differentiate among $n$ classes of landslide events. The classes may be
established using an absolute (A) or a relative (R) criterion, i.e., simply setting a minimum and
maximum number of landslides for each class or defining these numbers as landslide spatial
density, i.e. in terms of number of landslides per unit area. Six landslide density criteria have been
considered in the performed parametric analysis (**Table 8**) referring to the criteria used in the
Norwegian EWS (**Tab.1**). Two of them employ an absolute criterion using different numbers of
landslides per LE class the other four simulations, obtained considering the relative criterion, vary
as a function of both number of landslides and territorial extensions (10.000 $km^2$ and 15.000 $km^2$).
Changing the definition of LE classes, the duration matrix and the performance indicators vary
because of relocation of the $d_{ij}$ components. In particular the $time_{ij}$ element, which is the amount of
time for which a level i-[th] warning event is concomitant with a class j-[th] landslide event, may vary
the j-[th] index causing a movement of the element along the i-[th] row. The parametric analysis has
been performed using the landslide dataset A, which includes 385 landslides. **Table 9** reports the
classification of the LEs in the 6 combination of landslide density criteria.





**Tab. 8**. Parametric analysis: landslide density criteria considered to classify the LEs.

| LE class | Absolute criterion [No. of landslides] and number of LEs | | Relative criterion [No. of landslides / Area] and number of LEs | | | |
|---|---|---|---|---|---|---|
| | $A_{0,14}$ | $A_{1,18}$ | $R\text{-}15K_{0,14}$ | $R\text{-}15K_{0,10}$ | $R\text{-}10K_{0,14}$ | $R\text{-}10K_{0,10}$ |
| 0 | 0 | 1 | 0 | 0 | 0 | 0 |
| SMALL | 1 to 4 | 2 to 4 | (1 to 4)/15'000 km$^2$ | (1 to 4)/15'000 km$^2$ | (1 to 4)/10'000 km$^2$ | (1 to 4)/10'000 km$^2$ |
| INTERMEDIATE | 5 to 14 | 5 to 18 | ( 5 to 14)/15'000 km$^2$ | ( 5 to 10)/15'000 km$^2$ | ( 5 to 14)/10'000 km$^2$ | ( 5 to 10)/10'000 km$^2$ |
| LARGE | > 14 | > 18 | > 14/15'000 km$^2$ | > 10/15'000 km$^2$ | > 14/10'000 km$^2$ | > 10/10'000 km$^2$ |


**Tab 9.** Classification of LEs for the 6 simulations reported in table 8.

| LE class | Absolute criterion [No. of landslides] and number of LEs | | Relative criterion [No. of landslides / Area] and number of LEs | | | |
|---|---|---|---|---|---|---|
| | $A_{0,14}$ | $A_{1,18}$ | $R\text{-}15K_{0,14}$ | $R\text{-}15K_{0,10}$ | $R\text{-}10K_{0,14}$ | $R\text{-}10K_{0,10}$ |
| SMALL | 124 | 32 | 132 | 132 | 133 | 133 |
| INTERMEDIATE | 9 | 9 | 5 | 3 | 4 | 4 |
| LARGE | 4 | 4 | 0 | 2 | 0 | 0 |


As an example, the simulations $R\text{-}15K_{0,10}$ and $R\text{-}15K_{0,14}$ differ for the definition of both LE classes
Large and Intermediate. By comparing the two respoctive duration matrices (**Tab. 10-a; b**) a
movement of the durations from $d_{24}$ and $d_{34}$ to respectively $d_{23}$ and $d_{33}$ is evident. This behaviour is
due to the increase of spatial density for LE class Large, in particular from 0,67 landslides per 1000
km$^2$ to 0,93 landslides per 1000 km$^2$ (**Tab. 8**), which causes a relocation of time$_{i4}$ along the rows.
**Tab. 10.** Duration matrix results for simulations $R\text{-}15_{0,10}$ , $R\text{-}15_{0,14}$.

| $R\text{-}15K_{0,10}$ | | LE duration (h) | | | |
|---|---|---|---|---|---|
| | | 1 | 2 | 3 | 4 |
| | 1 | 600,48 | 107,62 | 0,00 | 0,00 |
| WE duration (h) | 2 | 9,88 | 8,47 | 0,98 | 0,82 |
| | 3 | 0,00 | 1,16 | 0,00 | 0,58 |
| | 4 | 0,00 | 0,00 | 0,00 | 0,00 |


| $R\text{-}15K_{0,14}$ | | LE duration (h) | | | |
|---|---|---|---|---|---|
| | | 1 | 2 | 3 | 4 |
| | 1 | 600,48 | 107,62 | 0,00 | 0,00 |
| WE duration (h) | 2 | 9,88 | 8,47 | 1,80 | 0,00 |
| | 3 | 0,00 | 1,16 | 0,58 | 0,00 |
| | 4 | 0,00 | 0,00 | 0,00 | 0,00 |




Changes within the duration matrix mean that the value of the performance indicators may change.
**Table 11** presents a summary of performance indicators for all six simulations of the landslide
density criteria used in the parametric analysis.

**Tab. 11:** Performance indicators for the six simulations of landslide density criteria considered in
the parametric analysis.

| Performance indicator | $A_{0,14}$ | $A_{1,18}$ | $R\text{-}15K_{0,14}$ | $R\text{-}15K_{0,10}$ | $R\text{-}10K_{0,14}$ | $R\text{-}10K_{0,10}$ |
|---|---|---|---|---|---|---|
| $I_{eff}$ | 0,95 | 0,86 | 0,98 | 0,98 | 0,98 | 0,98 |
| $HR_L$ | 0,95 | 0,86 | 0,99 | 0,99 | 0,99 | 0,99 |
| $PP_W$ | 1,00 | 1,00 | 0,99 | 0,99 | 0,99 | 0,99 |
| TS | 0,95 | 0,86 | 0,98 | 0,98 | 0,98 | 0,98 |
| OR | 18,98 | 6,07 | 42,75 | 42,75 | 49,43 | 49,43 |
| MR | 0,05 | 0,14 | 0,02 | 0,02 | 0,02 | 0,02 |
| $R_{MA}$ | 0,05 | 0,14 | 0,01 | 0,01 | 0,01 | 0,01 |
| $R_{FA}$ | 0,00 | 0,00 | 0,01 | 0,01 | 0,01 | 0,01 |
| ER | 0,05 | 0,14 | 0,02 | 0,02 | 0,02 | 0,02 |
| MFB | 1,00 | 1,00 | 0,61 | 0,61 | 0,55 | 0,55 |


The results show similar performance for the four simulations derived using a relative criterion
($R15\text{-}C_{0,14}$ $R15\text{-}C_{0,10}$ $R10\text{-}C_{0,14}$ $R10\text{-}C_{0,10}$) . The values of the success indicators are always high:
well above 95%, for $I_{eff}$, HR, TS, $PP_w$, while OR ranges between 42 and 49 (**Fig. 12.a**). This is due
to the high value of CPs compared to those of MAs and FAs, underlining a good performance of the
early warning model for these four simulations. In fact, also the error indicators are very low in
terms of percentage, around 1-2% (**Fig. 12.b**). Lower values are observed for the combination
obtained considering the absolute criterion, and in particular for $A_{1,18}$, with MR, $R_{MA}$ and ER
around 14%. The MFB is generally high for all simulations denoting a bad capability of the model
to predict LEs of classes 3 and 4. Anyway, it must be emphasized that, considering these landslide
density criteria, only the simulations $R\text{-}15K_{0,10,}$ $A_{0,14}$ and $A_{1,18}$ have LEs of class 4 in the period of
the analysis (**Tab. 8**).
In conclusion, the parametric analysis shows significant differences between the absolute and
relative criterion simulations. For this case study, absolute criterion simulations have lower success
performance indicators, in particular for the values of odds ratio (OR) and, very high values of
missed and false alert balance (MFB) compared to the performance indicators obtained for  relative





criterion simulations. Moreover, the absolute criterion simulations produce a number of purple
errors that increase the PSM (**Fig. 13.b**).

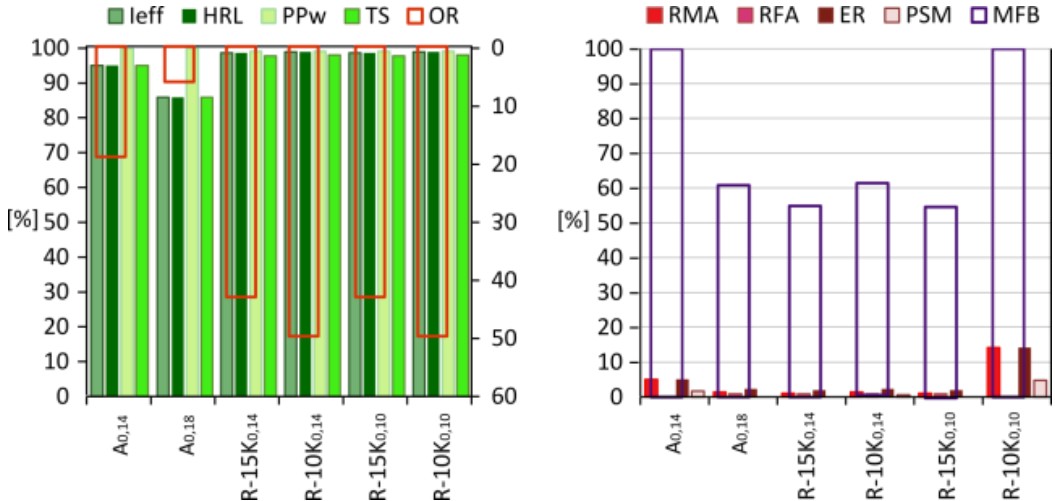

**Fig. 12**: Performance indicators related to the success (a) and to the errors (b) of the warning model,
evaluated for the six simulations of landslide density criteria considered in the parametric analysis.

## 5. Conclusions

The main aim of regional landslide early warning systems is to produce alert advices within a
specific warning zone and to inform local authorities and the public of landslide hazard at a given
level. To evaluate the performance of the alerts issued by such systems several aspects need to be
considered, such as: the possible occurrence of multiple landslides in the warning zone, the duration
of warnings in relation to the time of occurrence of landslides, the level of the issued warning in
relation to spatial density of landslides in the warning zone and the relative importance system
managers attribute to different types of errors. To solve these issues, the EDuMaP method can be
seen as a useful tool for testing the performance of regional landslide warning models. Up to now,
the method has been applied exclusively to systems that issue alerts on fixed warning zones. By
using data from the Norwegian landslide EWS this study has extended the applicability of the
EDuMaP method to warning systems that uses variable warning zones. In this study, the EDuMaP
method has been used to evaluate the performance of the Norwegian landslide early warning system
for Vestlandet (Western Norway) for the period 2013-2014. The results show an overall good
performance of the system for the area analyzed. Two datasets of landslide occurrences have been
used in this study: the first one including all the slope failures registered and gathered in the NVE



database within the test area; the second one excluding the phenomena whose typology was either
not determined or is not typically associated to rainfall. The results are not too sensitive to the
dataset of landslides, although slightly better results are registered with the smallest (i.e. more
accurate) dataset. In both cases, the high value of the MFB highlights a high number of MAs
compared to the FAs. A recommendation could be to have a MFB lower than 25%, which means
that only 1 wrong alert out of 4 is a MA. Following this reasoning, a reduction of the warning level
"High" is recommended in order to reduce the MAs and to increase the performance of the
Norwegian EWS.
A parametric analysis was also conducted for evaluating the performance sensitivity, to the
landslide density criterion, Lden(k), used as an input parameter with EDuMaP. This parameter
represents the way landslide events are differentiated in classes. In the analysis the classes were
established considering both absolute (2 simulations) and relative (4 simulations) criteria. The
parametric analysis shows how the variation of the intervals of the LE classes affects the model
performance. The best performance of the alerts issued in Western Norway was obtained applying a
relative density criterion for the definition of the LE classes. The parametric analysis shows only
minor differences in the performance analysis among the four cases considered with the relative
density criteria. In conclusion, this study highlights how the definition of the density criterion to be
used in defining the LE classes is a fundamental issue that system managers need to be take into
account in order to give an idea on the number of landslides expected for each warning level over a
given warning zone.

## 564 Acknowledgement

This work was carried out during a research period of LP as visiting PhD student at NVE, Oslo. The
authors are grateful to two NVE employees: Søren Boje for his criticism and comments and, Julio
Pereira for GIS data sharing.

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
