# Peer review of "Published: 16 January 2017"

_Natural Hazards and Earth System Sciences, 2017_

## Referee Comment (RC1) · Anonymous Referee #1 · 13 Feb 2017

General comment / remark:

The early warning system (EWS) in Norway described in this paper is based on real-time observation of hydro-meteorological condition, landslide occurrence, pre-defined hazard threshold levels, landslide inventory and susceptibility maps. The system provides daily regional alerts and warnings on landslide throughout the country to the public through website (http://www.varsom.no/en/). Its performance during the operation period from 2013 to 2014 was evaluated and the results indicated that the performance was generally good with high rate of correct prediction and low rate of false alarm or missed events. Room for improvement in operation has also been identified and proposed. This EWS can be a good reference/example for other parts of the world

where rainfall-induced landslide warning system is needed and respective datasets, viz. real-time rainfall and landslide observation, susceptibility maps, landslide inventory are present.

Specific comments:

1. Some figures are unclear and difficult to read. Please improve the legibility of the figures as far as possible.

2. Currently, the warning levels are updated twice per day. Given that heavy rainstorms can develop rapidly, suggest to update at shorter time interval in some situation such that appropriate warning levels can be issued in time before landslide occurrence.

3. Some tables and figures are incorrectly referred in the text (e.g. "Table 2" in line 427 should read Table 4). Suggest the author to review all table and figure numbers.

4. "R" in lines 168 and 173 should read "Red".

5. "Tab." and "Fig." through the manuscript should read "Table" and "Figure".

6. The "Probability of serious mistakes" as one of the performance indicators in Table 4 has not been evaluated in subsequent sessions.

---

## Referee Comment (RC2) · Anonymous Referee #2 · 13 Feb 2017

This manuscript assesses the performance of a national early warning system for regional landslide occurrence that was established recently in Norway. To this end, a performance-evaluation method EDuMaP (originally developed in Italy) was adapted to the case of Norway where spatial warning units are not constant but variable in space from case to case. While overall the landslide early warning system (LEWS) seems to perform quite well, this study also revealed that such a performance analysis strongly depends on the criterion selection.

Assessing the performance of such a country-wide LEWS is of great interest to NHESS readers as such warning systems are still new and not well-established yet. The manuscript provides a good description of the warning system and shows an interesting approach how it can be evaluated in a systematic manner. In that sense, I see a substantial potential for publication in this journal. On the other hand, I have a number of major questions and comments that, I think, deserve some further work:

1) The analyzed data set (both the number of warnings and observed landslides) is limited. It includes only warnings of three warning classes (green, yellow and orange) and a relatively low number of landslide observations (in particular for case B) without any landslide event classified as "large" (line 388). So one of my main question is: why was this analysis restricted to Vestlandet only and not performed for whole Norway? And why does it only include data from two years? I'm afraid that with this limitation (in particular with the missing of red warnings) the performance analysis is not comprehensive enough to draw strong conclusions.

2) Coming from another research field than "performance analysis" I had substantial difficulties to understand the extended EDuMaP-method (section 3.3). In particular, I was missing the "rationale" behind this method. In simple words: What's the rationale behind the assumption that an issued warning was successful or less successful. For example, is it more important that the location of an issued warning is correct than its intensity? Or is it most important that an warning is issued for day 1 even if the location and intensity is somewhat over- or underestimated? I suggest that the authors very clearly explain their rationale behind their technical assumptions.

3) I'm missing a benchmark for this performance evaluation. Is this landslide early warning system successful or not in comparison with other early warning systems worldwide? On lines 66 to 70 the authors mention a number of other such early warning systems – some of them are regional, others are local – and, in addition, there are also many flood early warning systems worldwide. I'm sure some of them have been evaluated in a similar way than this one. For the reader, it would be important to know (as a conclusion from this work) how the performance of this EWS compares with others.
4) Fig. 9b seems to omit the category "no warning issued – no event observed" while Fig. 9a seems to include this category (True Negatives). Is this mentioned somewhere? On what basis did you do this? As a result, the green category (in Fig. 9b) seems to be underrepresented. Yellow seems to dominate (but this is only for cases with either a warning or an observed landslide.) I think this gives different messages if you include or exclude the category "no warning issued – no event observed". From Fig. 9b the authors conclude that for Case B the EWS performs slightly better than for Case A. I would say the difference is very small . . . and I wouldn't over-interpret Fig. 9.

5) That brings me to another issue: is it really necessary (and of added value) to conduct the performance analysis for the two cases? Why don't you show only results for Case B (as you seem to distrust the data from Case A that you omit in Case B). Again – as stated above – I would suggest to extend the analysis to the entire country and to the entire period of the warning system, but exclude those landslide observations that you distrust.

6) The list of references includes many reports . . . some of them in Norwegian . . . please check which of these reports are really important for the understanding of this paper. (for example, do we really need all these references on geology and landforms?). On the other hand, I'm missing references to other authors (than Calvello and Piciullo) on performance evaluation of warning systems. There must be some of them!

Minor comments:

The abstract is not well balanced between introduction (background) and results (conclusions). There is too much background and introduction about the EWS. I suggest to shorten that substantially.

Line 47: "which are increasing with climate change"; I would say: "which are expected to increase with cc"

On line 75 the authors mention for the first time the fact that the Norwegian EWS issues

"variable" warning zones. It is very important that the authors clarify what they mean with "variable". I suggest to write "warning zones with a variable extent (or: area)".

Line 110: "In contrary" should be "On the other hand,"

Line 216: "are observed described"; either observed or described The authors use the term "precipitation episodes" several times in the text. I'm not sure "episodes" is the correct term here. I would rather suggest "events".

Line 254: "are shown" (not "are showed")

Table 3 is not necessary because all this information is given in the text already.

Line 331: "the some" should be "the same"

Line 335: "in Day 1" should be "on Day 1"

Line 335: "appears" should be "appear"

Tables 5 and 6 are not necessary because all this information is given in the text already

Line 427: "Tab.2" should be "Tab. 4"

I'm not sure all of the Figures are really needed. Please carefully reconsider which of Figs. 1 to 6 (on the EWS and its application) are really needed.

---

## Author Comment (AC1) · 10 Mar 2017

The Authors thank Reviewer #1 for his/her positive comment to our paper. The manuscript has been revised according to the specific comments provided. In particular all the figures have been improved in the revised manuscript.

---

## Author Comment (AC3) · 10 Mar 2017

table

[Figure]

| Warning levels | yellow | orange | red |
|----------------|--------|--------|-----|
| 2013 | 21 | 0 | 0 |
| 2014 | 34 | 5 | 0 |
| 2015 | 20 | 2 | 0 |
| 2016 | 21 | 0 | 0 |

**Fig. 1.**

---

## Author Response (AR1)

**Replies to comments of Reviewers #1 and #2**

Submission ID: nhess-2017-24

We would like to thank the Editor and the Reviewers for their careful review and their valuable comments, which have been constructive and useful to improve the quality of the manuscript.

Our replies to general and specific comments of Reviewers #1 and #2 are listed below.

**Anonymous Referee #1**

General comment / remark:

The early warning system (EWS) in Norway described in this paper is based on realtime observation of hydro-meteorological condition, landslide occurrence, pre-defined hazard threshold levels, landslide inventory and susceptibility maps. The system provides daily regional alerts and warnings on landslide throughout the country to the public through website (http://www.varsom.no/en/). Its performance during the operation period from 2013 to 2014 was evaluated and the results indicated that the performance was generally good with high rate of correct prediction and low rate of false alarm or missed events. Room for improvement in operation has also been identified and proposed. This EWS can be a good reference/example for other parts of the world where rainfall-induced landslide warning system is needed and respective datasets, viz. real-time rainfall and landslide observation, susceptibility maps, landslide inventory are present.

R: We thank Reviewer #1 for his/her positive comment to our paper.

Specific comments:

1. Some figures are unclear and difficult to read. Please improve the legibility of the figures as far as possible.

R: According to the comment, figures 4, 5, have been updated and improved as suggested. Thank you.

2. Currently, the warning levels are updated twice per day. Given that heavy rainstorms can develop rapidly, suggest to update at shorter time interval in some situation such that appropriate warning levels can be issued in time before landslide occurrence.

R: The warning levels are updated minimum twice per day based on weather forecast. The system manager, NVE, receives weather forecast updates 4 times per day and, using this information, sends the warnings as early as possible from 66 hours to few hours ahead. This information was added to the manuscript to better clarify this point. Thank you.

3. Some tables and figures are incorrectly referred in the text (e.g. "Table 2" in line 427 should read Table 4). Suggest the author to review all table and figure numbers.

R: We checked all the figures and tables. Thank you.

4. "R" in lines 168 and 173 should read "Red".

R: We modified as suggested. Thank you.

5. "Tab." and "Fig." through the manuscript should read "Table" and "Figure".

R: According to the comment, we modified "Tab." and "Fig." through the manuscript. Thank you.

6. The "Probability of serious mistakes" as one of the performance indicators in Table 4 has not been evaluated in subsequent sessions.

R: We thank the Reviewer for his/her comment, however the "Probability of serious mistakes" has been evaluated in the performance analysis but was erroneously omitted in table 11. Figure 12 was also revised due to a different error we found in the position of the bars.

**Anonymous Referee #2**

General comment / remark:

This manuscript assesses the performance of a national early warning system for regional landslide occurrence that was established recently in Norway. To this end, a performance-evaluation method EDuMaP (originally developed in Italy) was adapted to the case of Norway where spatial warning units are not constant but variable in space from case to case. While overall the landslide early warning system (LEWS) seems to perform quite well, this study also revealed that such a performance analysis strongly depends on the criterion selection. Assessing the performance of such a country-wide LEWS is of great interest to NHESS readers as such warning systems are still new and not well-established yet. The manuscript provides a good description of the warning system and shows an interesting approach how it can be evaluated in a systematic manner. In that sense, I see a substantial potential for publication in this journal.

R: We thank Reviewer #2 for his/her interest in our manuscript. We carefully revised the manuscript according to the many valuable comments and recommendations provided by both Reviewers.

On the other hand, I have a number of major questions and comments that, I think, deserve some further work:

1) The analyzed data set (both the number of warnings and observed landslides) is limited. It includes only warnings of three warning classes (green, yellow and orange) and a relatively low number of landslide observations (in particular for case B) without any landslide event classified as "large" (line 388). So one of my main question is: why was this analysis restricted to Vestlandet only and not performed for whole Norway? And why does it only include data from two years? I'm afraid that with this limitation (in particular with the missing of red warnings) the performance analysis is not comprehensive enough to draw strong conclusions.

R: We thank Reviewer #2 for this comment that gives us the possibility to better explain the reasons of choosing this case study and dataset. The "Vestlandet" region was chosen as it is one of the areas most prone to landslides in Norway. Moreover, for this area the landslide database is more reliable and complete than in the rest of Norway. As the second most populated area of the Nation, more information on landslides are available.

The Norwegian national landslide early warning system (LEWS) is a realtively new system that became operational in 2013. The analyses presented in this manuscript started in 2015 and only data for 2013-2014 were available at that time. A large work of collection and checking of landslide information from different sources (NVE, rails and roads Authority, other databases, media) was carried out, with the aim of avoiding repetitions and providing a reliable dataset. However, to answer this comment, we checked the number of warning issued in 2015-2016 in Vestlandet. There were only few days with Orange warnings and no one with Red warnings. The table below shows the number of warnings and the warning levels issued in Vestlandet in the period 2013-2016.

| Warning levels | yellow | orange | red |
|:---:|:---:|:---:|:---:|
| 2013 | 21 | 0 | 0 |
| 2014 | 34 | 5 | 0 |
| 2015 | 20 | 2 | 0 |
| 2016 | 21 | 0 | 0 |

As shown, the red level would still be missing even if we considered the period 2015-2016. According to the meaning of warning levels presented at http://www.varsom.no/en, the red level defines "an extreme situation that occurs very rarely, it requires immediate attention and may cause severe damages within a large extent of the warning area". Concluding, incorporating these data would not change the results of the performance analysis and would not add anything significant towards the main aim of the paper, i.e. proposing an extension of the EDuMaP method for the performance evaluation of LEWSs issuing warnings over zones characterised by a variable size. Finally the title of the paper has been modified in "*Adapting the EDuMaP method to test the performance of the Norwegian early warning system for weather-induced landslides*", for better clarifying the aim of the paper and avoiding confusion in the reader.

2) Coming from another research field than "performance analysis" I had substantial difficulties to understand the extended EDuMaP-method (section 3.3). In particular, I was missing the "rationale" behind this method. In simple words: What's the rationale behind the assumption that an issued warning was successful or less successful. For example, is it more important that the location of an issued warning is correct than its intensity? Or is it most important that an warning is issued for day 1 even if the location and intensity is somewhat over- or underestimated? I suggest that the authors very clearly explain their rationale behind their technical assumptions.

R: The EDuMaP method comprises three successive steps: identification and analysis of landslide and warning Events (E), from available databases; definition and computation of a Duration Matrix (DuMa), and evaluation of the early warning model Performance (P) by means of performance criteria and indicators. The parameters needed to carry on the events analysis (E) are ten. Among them, there is the spatial discretization adopted for warnings, $\Delta A(k)$, which describes if the warning zone is fixed or variable. For instance, the LEWS employed in Rio de Janeiro considers fixed warning zones, on the contrary the system adopted in Norway uses variable warning zones. In earlier studies, the EDuMaP method has been applied to analyse the performance of regional landslide EWSs adopting a fixed spatial discretization for warnings. When the landslide EWS employs variable warning zones, this characteristic significantly influences the first two steps of the EDuMaP method.

Section 3.3 was rewritten for increasing the comprehensibility of the methodology. It explains how to define landslide events (LEs) and warning events (WEs) and how to compute the duration matrix in case of variable warning zones. The landslides are grouped in LEs as a function of the warning zone in which they occur. A warning zone can be seen as an area alerted with the same level of warning (i.e., green, yellow, orange, red). The EDuMaP method evaluates the duration of each level of warning (i.e., green, yellow, orange, red) and the class of landslide event (i.e: the number of landslides) occurred over the time in a warning zone. In the EDuMaP method, a warning can be considered successful as a function of both the level of warning issued and the number of landslide occurred in the zone alerted. The number of landslides expected for each warning level often is defined by the LEWS managers, otherwise can be evaluated considering a landslide density criterion, $L_{den(k)}$.

3) I'm missing a benchmark for this performance evaluation. Is this landslide early warning system successful or not in comparison with other early warning systems worldwide? On lines 66 to 70 the authors mention a number of other such early warning systems – some of them are regional, others are local – and, in addition, there are also many flood early warning systems worldwide. I'm sure some of them have been evaluated in a similar way than this one. For the reader, it would be important to know (as a conclusion from this work) how the performance of this EWS compares with others.

R: Among LEWSs at a regional scale, the performance of the system is evaluated principally by computing the joint frequency distribution of landslides and alerts. Empirical evaluations are often carried out by simply analyzing the time frames during which significant high-consequence landslides occurred in the test area (Keefer et al., 1987; Aleotti, 2004; Cheung et al., 2006; Baum and Godt, 2010; Capparelli and Tiranti, 2010). Alternatively, the performance evaluation is based on 2 by 2 contingency tables computed for the joint frequency distribution of landslides and alerts, both considered as dichotomous variables (Yu et al., 2003; Cheung et al., 2006; Godt et al., 2006; Restrepo et al., 2008; Tiranti and Rabuffetti, 2010; Kirschbaum et al., 2012; Martelloni et al., 2012; Peres and Cancelliere, 2012; Staley et al., 2013; Lagomarsino et al., 2013, 2015; Greco et al., 2013; Segoni et al., 2014; Gariano et al., 2015; Stähli et al., 2015). The performance of the systems operational in Norway and Rio de Janeiro was analysed applying the EDuMaP method considering: the possible occurrence of multiple landslides in the warning zone; the duration of the warnings in relation to the time of occurrence of the landslides; the level of the issued warning in relation to the landslide spatial density in the warning zone; the relative importance system managers attribute to different types of errors.

In general it's difficult to compare the performance of LEWSs, especially if it has been evaluated with different methods. The values to evaluate the statistical indicators derive from different reasoning, for example, on what is considered as false, missed or correct alerts. Substantial differences may be observed among a 2x2 contingency table and a $n_x m$ duration matrix. The latter compares the $n$ levels of warning in relation to the $m$ classes of landslide events. The EDuMaP method evaluates the performance of a LEWS considering the number of warning levels and the classes of landslide events, thus, warnings and landslides are not considered as dichotomous variables as it is for contingency tables.

A benchmark could be defined, but it would require a separate analysis and a comparison of a relatively high number of different LEWSs evaluated with the EDuMaP method. Because system managers of LEWSs may attribute a relative importance to different aspects (i.e.: missed alerts, false alerts, purple errors, correct alerts, greens, the level of warning issued, classes of landslide, etc..). As a consequence, different performance criteria are needed to be chosen in order to consider the system managers choices and to carry on the performance analysis. Currently the authors are still working on a comparison among the performance evaluation of different LEWSs in order to provide "functioning standards".

4) Fig. 9b seems to omit the category "no warning issued – no event observed" while Fig. 9a seems to include this category (True Negatives). Is this mentioned somewhere? On what basis did you do this? As a result, the green category (in Fig. 9b) seems to be underrepresented. Yellow seems to dominate (but this is only for cases with either a warning or an observed landslide.) I think this gives different messages if you include or exclude the category "no warning issued – no event observed". From Fig. 9b the authors conclude that for Case B the EWS performs slightly better than for Case A. I would say the difference is very small . . . and I wouldn't over-interpret Fig. 9.

R: We thank the Reviewer for giving us the possibility to clarify some important concepts of the duration matrix, that erroneously we have neglected to mention in the manuscript. The component d11 ("no warning issued – no event observed") of the matrix expresses the number of hours when no warnings are issued and no landslides occur. Both criteria (1 and 2) purposefully neglect element d11, whose value is typically orders of magnitude higher than the values of the other elements of the matrix because it also includes all hours without rainfall, for which a LEWS is not designed to deal with, specifically. Thus, d11 component is neglected in our analysis in order to avoid an overestimation of the performance and to allow a more useful relative assessment of the information located in the remaining part of the duration matrix. So, in figure 9 a, b (currently figure 6 a,b) the d11 component of the duration matrix is neglected.

According to the suggestion provided we have modified the description for figure 9. Here are the new sentences: "In terms of criterion 2, Case B shows slightly higher values of Green (14%) than

Case A (7%). This means that considering the reduced set of landslides (Set b), there is a slightly better correspondence between the LE classes and corresponding warning levels issued". However, it doesn't mean a better performance for Case B, because figure 9 (currently figure 6) shows only preliminary results. With the EDuMaP method the performance is evaluated through the evaluation of statistical indicators (fig. 12 and tab. 11- currently fig. 9 and tab. 9).

5) That brings me to another issue: is it really necessary (and of added value) to conduct the performance analysis for the two cases? Why don't you show only results for Case B (as you seem to distrust the data from Case A that you omit in Case B). Again – as stated above – I would suggest to extend the analysis to the entire country and to the entire period of the warning system, but exclude those landslide observations that you distrust.

R: The dataset B is composed by a catalogue of landslides with a known typology. On the contrary the dataset A includes also landslides in soil of unknown typology that can be, anyway, classified as rainfall-induced landslides. For this reason we decided to keep both the datasets. Finally, the results coming from the two datasets were compared to evaluate the differences in terms of performance indicators arising from uncertainties in the landslide database.

6) The list of references includes many reports . . . some of them in Norwegian . . . please check which of these reports are really important for the understanding of this paper. (for example, do we really need all these references on geology and landforms?). On the other hand, I'm missing references to other authors (than Calvello and Piciullo) on performance evaluation of warning systems. There must be some of them!

R: According to the suggestion all the references in Norwegian have been cancelled because considered not useful to improve the comprehension of the manuscript.

In literature two main approaches can be distinguished for the evaluation of the performance of LEWSs at a regional scale: empirical evaluations and 2x2 contingency tables. As already mentioned in the answer to comment No. 3, the firsts are often carried out by simply analyzing the time frames during which significant high-consequence landslides occurred in the test area (Keefer et al., 1987; Aleotti, 2004; Cheung et al., 2006; Baum and Godt, 2010; Capparelli and Tiranti, 2010). The latter are computed for the joint frequency distribution of landslides and alerts, both considered as dichotomous variables (Yu et al., 2003; Cheung et al., 2006; Godt et al., 2006; Restrepo et al., 2008; Tiranti and Rabuffetti, 2010; Kirschbaum et al., 2012; Martelloni et al., 2012; Peres and Cancelliere, 2012; Staley et al., 2013; Lagomarsino et al., 2013, 2015; Greco et al., 2013; Segoni et al., 2014; Gariano et al., 2015; Stähli et al., 2015). The EDuMaP method is a different approach taking into account: the possible occurrence of multiple landslides in the warning zone, the duration of the warnings in relation to the time of occurrence of the landslides, the level of the issued warning in relation to the landslide spatial density in the warning zone and the relative importance system managers attribute to different types of errors. A comparison between the EDuMaP method and other methodologies for the evaluation of the performance lies outside the scope of the paper, which is focused on the definition of an original approach, to be implemented in the EDuMaP method, for the computation of the elements of the duration matrix in the case of early warning models issuing alerts on variable warning zones. Many references to different approaches for the performance evaluation were presented in Calvello and Piciullo 2016, and Piciullo et al., 2016.

Minor comments:

The abstract is not well balanced between introduction (background) and results (conclusions). There is too much background and introduction about the EWS. I suggest to shorten that substantially.

R: According to the suggestion, the abstract has been rewritten. The description of the Norwegian LEWS was too long and has been shortened. Now is less than half compared to the introduction and conclusions. Thank you.

Line 47: "which are increasing with climate change"; I would say: "which are expected to increase with cc"

R: We modified as suggested.

On line 75 the authors mention for the first time the fact that the Norwegian EWS issues "variable" warning zones. It is very important that the authors clarify what they mean with "variable". I suggest to write "warning zones with a variable extent (or: area)".

R: The sentence has been modified in:" Daily alerts are issued throughout the country in variable size warning zones".

Line 110: "In contrary" should be "On the other hand,"

R: Corrected as suggested.

Line 216: "are observed described"; either observed or described

R: observed has been deleted.

The authors use the term "precipitation episodes" several times in the text. I'm not sure "episodes" is the correct term here. I would rather suggest "events".

R: We changed in "precipitation events". Thank you

Line 254: "are shown" (not "are showed")

R: Corrected. Thank You

Table 3 is not necessary because all this information is given in the text already.

R: Table 3 was cancelled.

Line 331: "the some" should be "the same"

R: Corrected. Thank You

Line 335: "in Day 1" should be "on Day 1"

R: Corrected. Thank You

Line 335: "appears" should be "appear"

R: Corrected. Thank You

Tables 5 and 6 are not necessary because all this information is given in the text already

R: Table 5 was cancelled whereas table 6 is useful to summarize all the information on number of landslides, landslides, warning events issued and warning zones alerted in 2013-2014 in the area of analysis.

Line 427: "Tab.2" should be "Tab. 4"

R: Modified. Thank You

I'm not sure all of the Figures are really needed. Please carefully reconsider which of Figs. 1 to 6 (on the EWS and its application) are really needed.

R: We accepted the comment and decided to cancel figures 1, 3 and 6 because judged as not useful to fulfill the main aim of the paper. Thank you

[revised manuscript text omitted]

---

## Author Response (AR2)

**Replies to comments of Reviewers #2**

Submission ID: nhess-2017-24

**Anonymous Referee #2**

I have noted that the authors carefully considered my previous comments and comprehensively revised the manuscript. Nevertheless, I must say that not all of my major concerns have been fully resolved. For example, I still think that the data set is critically limited (without any red warning).

The Norwegian national landslide early warning system (LEWS) is a relatively new system operational since 2013. The analyses presented in this manuscript started in 2015 and only data for 2013-2014 were available. A large work of collection and checking of landslide information from different sources (NVE, rails and roads Authority, other databases, media) was carried out, with the aim of avoiding repetitions and providing a reliable dataset. For this reasons the study area was restricted to Vestlandet. In this area, a reliable dataset of landslides for the years 2015-2016 is still not available. Nevertheless, we checked the number of warnings issued in the last two years (2015-2016) in Vestlandet and no Red warnings were issued. The table below shows the number of warnings and the warning levels issued in Vestlandet in the period 2013-2016.

| Warning levels | yellow | orange | red |
|:---:|:---:|:---:|:---:|
| **2013** | 21 | 0 | 0 |
| **2014** | 34 | 5 | 0 |
| **2015** | 20 | 2 | 0 |
| **2016** | 21 | 0 | 0 |

As shown, the red level would still be missing even if we considered the period 2013-2016. According to the meaning of warning levels presented at http://www.varsom.no/en, the red level defines "an extreme situation that occurs very rarely, it requires immediate attention and may cause severe damages within a large extent of the warning area". Concluding, incorporating data from the years 2015-2016 would not change the results of the performance analysis and would not add anything significant towards the main aim of the paper, i.e. proposing an extension of the EDuMaP method for the performance evaluation of LEWSs issuing warnings over variable size zones.

And I still miss the rationale behind the assumptions of the EDuMaP method.

LEWSs may adopt a fixed or a variable spatial discretization for warnings ($\Delta A_{(k)}$).

In the first case the warning zones are univocally defined with fixed extents. For each warning zone, the warnings are issued over the whole zone according to site specific rainfall thresholds and decisional algorithms. Thus, only one level of warning can be issued in each warning zone in the minimum temporal discretization adopted for warnings ($\Delta t$). The performance analysis with the EDuMaP method is carried out separately for each warning zone. Therefore, in this case, the dij components of the duration matrix represent the time evaluation of the combination of warning levels issued and landslide events occurred in a specific warning zone in a period of analysis.

In the case of a variable spatial discretization for warnings the number and extent of the warning zones vary in time in the period of analysis ($\Delta T$). The number of warning zones is defined by the number of warning levels issued in the minimum temporal discretization ($\Delta t$). For instance, if only two levels (e.g. green and orange) are issued in a given $\Delta t$, the area of analysis (A) would be divided into two warning zones. The extent of the warning zones is obtained grouping together all the territorial units alerted with the same level of warning. In this paper, the territorial units are defined looking at the administrative municipal boundaries. In a given $\Delta t$, the Event analysis phase is carried out for all the warning zones simultaneously. The time evaluation of the elements of the duration matrix in a given $\Delta t$ ($time_{ij}$) for the area of analysis (A) is carried out by weighting the spatial contribution of each warning zone in relation to the total area, as follows: $time_{ij} = \Delta t * TU_{ij}/A$, where $TU_{ij}$ is the extent of the territorial units alerted with a warning level i and class of landslide event j in a given $\Delta t$.

In earlier studies, the EDuMaP method has been applied to analyse the performance of regional landslide EWSs adopting a fixed spatial discretization for warnings (i.e. fixed warning zones). In contrast, the Norwegian landslide EWS employs variable size warning zones which influence, as explained, the definition of the first two phases of the EDuMaP method: identification of landslide and warning events from available databases; definition and computation of the duration matrix. In this last revised version of the manuscript, the text better explains how to define LEs and WEs and how to compute the duration matrix in case of variable size warning zones. A reorganization of sections was carried out in order to increase the comprehension of the method. Section 3 now describes the EDuMaP method and how it was adapted for variable warning zones. The definition of the area of analysis and the description of the available datasets have been moved in section 4.

Still I'm convinced that this manuscript could become a valuable contribution for NHESS. At this stage I don't have any further specific suggestions for revision.

We thank you the reviewer for his comments.